METHODS

# Differential quantification of alternative splicing events on spliced pangenome graphs

**Simone Ciccolella**[1�], **Davide Cozzi**[1�], **Gianluca Della Vedova**[1], **Stephen Njuguna Kuria**[2], **Paola Bonizzoni**[1‡], **Luca Denti** [1,3‡]*

**1** Department of Computer Science, University of Milano-Bicocca, Milan, Italy, **2** Department of Biochemistry and Biotechnology, Pwani University, Kilifi, Kenya, **3** Department of Applied Informatics, Faculty of Mathematics, Physics and Informatics, Comenius University in Bratislava, Bratislava, Slovakia

l These authors contributed equally to this work.
‡These authors also contributed equally to this work.
* luca.denti@unimib.it

**Data Availability Statement:** pantas is open source and publicly available at https://www.github.com/algolab/pantas. The repository also

## Abstract

Pangenomes are becoming a powerful framework to perform many bioinformatics analyses taking into account the genetic variability of a population, thus reducing the bias introduced by a single reference genome. With the wider diffusion of pangenomes, integrating genetic variability with transcriptome diversity is becoming a natural extension that demands specific methods for its exploration. In this work, we extend the notion of spliced pangenomes to that of *annotated spliced pangenomes*; this allows us to introduce a formal definition of Alternative Splicing (AS) events on a graph structure. To investigate the usage of graph pangenomes for the quantification of AS events across conditions, we developed `pantas`, the first pangenomic method for the detection and differential analysis of AS events from short RNA-Seq reads. A comparison with state-of-the-art linear reference-based approaches proves that `pantas` achieves competitive accuracy, making spliced pangenomes effective for conducting AS events quantification and opening future directions for the analysis of population-based transcriptomes.

## Author summary

The ever increasing availability of complete genomes is advancing our comprehension of many biological mechanisms and is enhancing the knowledge we can extract from sequencing data. Pangenome graphs are a convenient way to represent multiple genomes and the genetic variability within a population. Integrating genetic variability with transcriptome diversity can improve our understanding of alternative splicing, a regulation mechanism which allows a single gene to code for multiple proteins. However, many unanswered questions are limiting our comprehension of the relationship between genetic and trancriptomic variations. With this work, we start to fill this gap by introducing `pantas`, the first approach based on pangenome graphs for the detection and differential quantification of alternative splicing events. A comparison with state-of-the-art approaches based on linear genome prove that pangenome graphs can be effectively used to perform such an analysis. By integrating genetic and transcriptome variability in a

contains all the scripts to reproduce the experimental evaluations described in the manuscript. All data used in our experimental evaluation is publicly available. Annotated splicing pangenome graphs are available at https://zenodo.org/records/13740102. Instructions on how to download the data, preprocess them, and run the analyses are available at https://github.com/algolab/pantas?tab=readme-ov-file#experiments.

**Funding:** This research work has received funding from the European Union's Horizon 2020 research and innovation programme under the Marie Skłodowska-Curie grant agreement (PANGAIA No. 872539 and ITN ALPACA N.956229 to PB). The project is also supported by the grant MIUR 2022YRB97K, PINC, Pangenome Informatics: from Theory to Applications (PB). This work has received funding from the European Union's Horizon programme under the Horizon Europe grant agreement (ASVA-CGR No. 101180581 to LD). The funders had no role in study design, data collection and analysis, decision to publish, or preparation of the manuscript.

**Competing interests:** The authors have declared that no competing interests exist.

single structure, `pantas` can pave the way to next generation bioinformatic approaches for the accurate analysis of the relations between genetic variations and alternative splicing.

## Introduction

Pangenomics is emerging as a new powerful computational framework for analyzing the genetic variability of a population without being negatively affected by the reference bias introduced when considering a single genome as a reference. The recent release of the first draft human pangenome reference [1] demonstrated that this richer reference can be used to perform many bioinformatics analyses, e.g. variant calling, with superior accuracy and precision, especially in structurally complex loci of the genome. In particular, pangenomes have also been showed to improve transcriptomic data analysis [2], thanks to the extension of the notion of pangenomes to that of spliced pangenomes (or pantranscriptomes). A spliced pangenome is obtained by enriching a pangenome, which is a graph structure representing the genetic diversity of multiple individuals of a population, with transcript variability coming from gene annotations. This complex but powerful graph structure is used to perform haplotype-aware transcript quantification [2]. A comparison with reference-based approaches proves that spliced pangenomes improve the accuracy of transcript quantification. Indeed, by incorporating both genetic variability and isoform diversity, spliced pangenomes allow to reduce the allelic bias and increase the number of reads aligned over heterozygous variations [2]. Apart from this seminal work, the usage of spliced pangenomes in the context of transcriptomic is fairly unexplored and their full potential is still under investigation. Similarly to pangenomics, pantranscriptomics needs suitable bioinformatics tools to fulfill its biological potential. Encouraged by the results in [2], we explore the adoption of spliced pangenomes for performing a classical transcriptomic task, namely the detection and differential quantification of Alternative Splicing (AS) events across conditions.

Alternative splicing (AS) is a regulation mechanism which allows a single gene to code for multiple isoforms and express multiple proteins. Such a mechanism is the main contributing factor for the overwhelming complexity of transcriptomes in eukaryotes. Alternative splicing is associated with different diseases [3–5] and it also plays a key role in various biological processes, like aging [6]. The advent of RNA-Sequencing technology (RNA-Seq) enabled the analysis of transcriptome and alternative splicing at an unprecedented speed and precision. A typical RNA-Seq analysis consists of comparing two conditions (e.g., control vs tumor) and checking for changes in terms of isoform abundances [7, 8]: a change in the relative abundances of the isoforms of a gene is usually a strong evidence of differential splicing. An alternative approach consists of directly detecting and quantifying alternative splicing events. Instead of focusing on entire isoforms quantification, several approaches work at a finer-grained level and focus on exon-exon boundaries, also known as splice junctions, and check for changes in their usage. By analyzing splice junctions, these approaches can directly detect and quantify AS events providing more accurate results with respect to approaches based on transcript quantification [9]. Several tools have been proposed in the literature to differentially quantify AS events from RNA-Seq datasets [10–17].

Differently from the state-of-the-art, where a single reference is considered and AS event quantification is modeled without taking into account the genetic variability of the population of the species under investigation, we propose to use a spliced pangenome and quantify AS events using this more powerful structure that inherently represents multiple individuals and

gene annotations. To this aim, we introduce the notion of *annotated spliced pangenome*, we propose the first formalization of AS events on this richer structure, and we provide the first method, `pantas`, to perform AS events differential quantification on a spliced pangenome. Although the adoption of a graph structure is not new in computational transcriptomics, where *splicing graphs* are commonly used to model isoform variability [11–13, 18–20], no approach takes also into account the genetic variability of the population under investigation. It is indeed well known that genetic variations alter splicing pattern in many diseases [21, 22]. However, establishing their impact on alternative splicing is still challenging [23]. Even though the adoption of spliced pangenomes may shed more light on the relationship between genetic variations and splicing, the current lack of efficient tools for pangenomic-based analysis of RNA-Seq data is hindering this kind of investigation. Our work here is a first step in closing this gap.

Experimental evaluation on simulated and real data from Drosophila and Human shows that spliced pangenomes can be effectively adopted to perform AS events detection and quantification. A comparison with state-of-the-art tools based on linear reference or splicing graphs shows that AS event quantification from spliced pangenome provides competitive results. Our preliminary investigation paves the way to the adoption of spliced pangenomes for pantranscriptomic analysis, where genes are annotated w.r.t. a pangenome and not a single reference genome [24].

## Methods

The problem we want to tackle is detecting and quantifying alternative splicing events supported by an input RNA-Seq dataset comprising two conditions with optional replicates with respect to a spliced pangenome. To this aim, we introduce the notion of *annotated spliced pangenome* and we describe how AS events can be detected from this graph structure. Finally, we present the `pantas` tool we developed to detect and differentially quantify AS events. From a high-level point of view, `pantas` is a set of utilities that helps the user to perform any step required for alternative splicing events quantification on spliced pangenome graphs. `pantas` builds and annotates a spliced pangenome starting from standard file formats (more details in Appendix A in S1 Text) and, after RNA-Seq reads have been aligned to the graph using state-of-the-art spliced aligners, it performs AS events detection and quantification against this richer structure. `pantas` detection can be summarized as follows:

1. annotated spliced pangenome is augmented with alignment information.

2. events are detected by analyzing the graph and applying the definitions provided below.

3. events are differentially quantified by comparing the conditions.

   More details on `pantas` implementation are available at the end of this section.

### Annotated spliced pangenome graph

A *variation graph*, as defined in [25], is a directed graph whose vertices correspond to portions of genomes and are labeled by nonempty strings, and edges represent consecutiveness between two such portions of genomes. A *spliced pangenome*, introduced in [2], is a variation graph with some distinguished walks. A *walk* in a graph is a sequence of vertices connected by edges. In a spliced pangenome, we can identify the walks $\{H_1, H_2, \ldots, H_h\}$ representing the haplotypes and the walks $\{T_1, T_2, \ldots, T_t\}$ representing the haplotype-aware transcripts. Although spliced pangenome can contain cycle, in this work we will focus on acyclic graphs. Each haplotype is the genomic sequence of a particular individual. We note that this sequence is usually a set of

sequences (one per chromosome). However, without loss of generality, we will consider the haplotype walks as single walks in the graph. Each haplotype-aware transcript is a transcript, i.e., a sequence of the coding regions of a gene, known as *exons*, that belongs to a specific haplotype in $\{H_1, \ldots, H_h\}$. We note that due to haplotype similarity, two haplotype-aware transcripts can be the same sequence of vertices of the graph, i.e., the same walk. The edges that belong to some transcript walks but do not belong to any haplotype walk correspond to annotated *splice junctions*. These edges link two vertices of the graph that are not consecutive in the haplotype walk due to the presence of intronic regions (non-coding regions) between the exons. Notice that, in a spliced pangenome, a vertex $v$ that belongs to at least a haplotype-aware transcript walk represents an exonic region. All other vertices represent intronic or intergenic regions that belong to some haplotype.

An *annotated spliced pangenome* is a spliced pangenome whose vertices and edges are annotated (or labeled) with additional information that will be essential for formalizing and detecting AS events. An example of *annotated spliced pangenome* is given in Fig 1. More precisely, each vertex $v$ of an annotated spliced pangenome is annotated with two information: the set $\mathcal{E}(v)$ of its exon labels (from now on simply exons) and the set $\mathcal{T}(v)$ of its haplotype-aware transcripts. We note that an exon is univocally identified by the haplotype-aware transcript it belongs to and by its order along that transcript. We note that a vertex can represent a single exonic region but, in an annotated spliced pangenome, this region may belong to multiple exons coming from different haplotype-aware transcripts, e.g., due to haplotype similarity

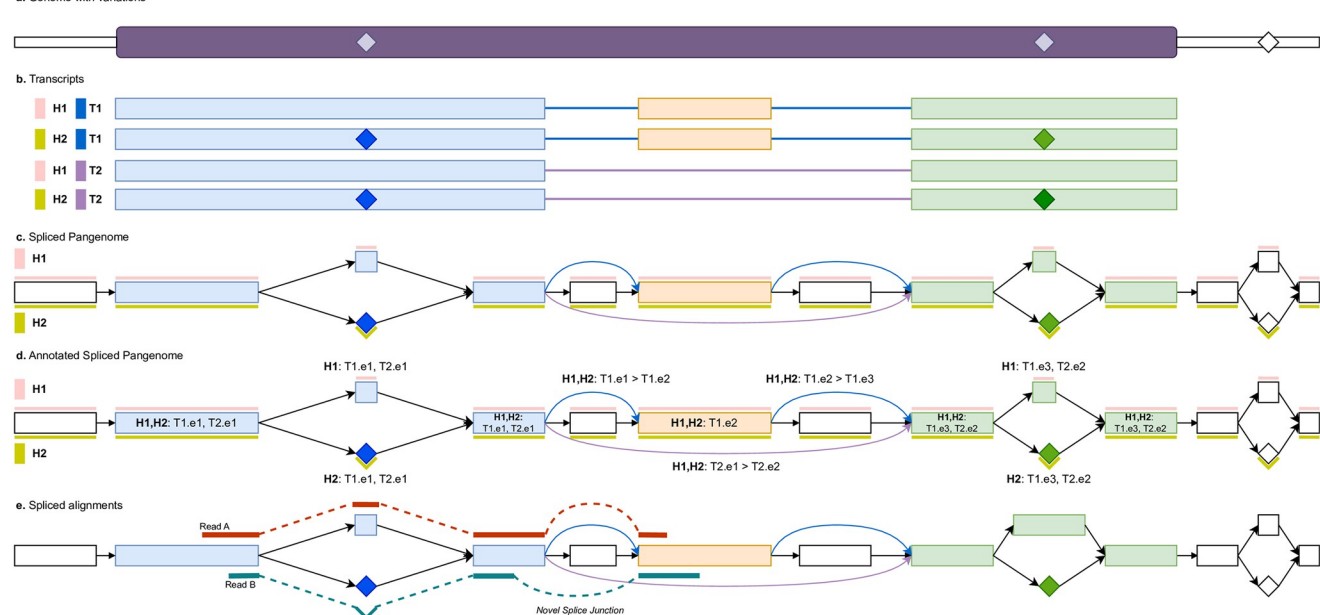

**Fig 1. Annotated spliced pangenome example.** (a) Reference genome with a purple box representing a gene locus and diamonds representing alternate alleles of variations. In this example, we will consider two haplotypes, H1 containing the reference alleles and H2 containing the alternate alleles. (b) The gene has two transcripts, namely T1 and T2, expressing an exon skipping event. There is a total of 4 haplotype-aware transcripts. The alternate alleles are color-coded depending on the exon they fall in. (c) Spliced pangenome built from the reference genome, gene annotation, and set of variations. Colored vertices and edges belongs to the transcript walks and they represent portions of exons and splice junctions. White vertices are genomic portions coming from intron and intergenic regions that do not belong to the transcript walks. The two haplotypes are represented as colored bars above and below the vertices. (d) Annotated spliced pangenome where each colored vertex (exon portion) is annotated by the transcripts and exons it belongs to and each colored edge (junction) is annotated with the junction information. (e) Spliced alignments of two RNA-Seq reads to the annotated spliced pangenome. Read A aligns over the reference allele of the first variation and supports the annotated splice junction between the first and second exons (blue edge). Read B, instead, aligns over the alternate allele of the variations and then supports a novel splice junction between the two exons (the novel junction induces an alternative donor events).

or to an alternative form of an exon in the case of an alternative splicing event. For this reason, $\mathcal{E}(v)$ is a set of exons and not a single exon. To simplify the exposition, whenever $\mathcal{E}(v)$ or $\mathcal{T}(v)$ are empty, we say that the vertex $v$ has no annotation—this can happen, e.g., for a vertex in an intronic or intergenic region. Moreover, each edge $e = \langle u, v \rangle$ of an annotated spliced pangenome that represents a splice junction is annotated with the set $\mathcal{T}(e)$ of the haplotype-aware transcripts it belongs to. As for vertices, the same splice junction can be shared by many haplotype-aware transcripts. In addition to this information that is required to fully represent the known haplotype-aware transcripts encoded in a spliced pangenome, the elements of an annotated spliced pangenome can be further annotated with additional data. For instance, following a step of RNA-Seq read alignment, each edge $e = \langle u, v \rangle$ of the graph can be annotated with a numeric value $w(e)$, representing its support (or coverage).

## Criteria for alternative splicing event detection

In this section we will provide the formal description of how alternative splicing events can be detected from an annotated spliced pangenome. We will first focus on annotated events and then we will formalize novel events. From a high level point of view, alternative splicing events can be identified by locally comparing the splice junctions of two transcripts represented in the annotated spliced pangenome. Thus pantas analyzes two sets of junctions, one coming from the *canonical* transcript whereas the other from the *alternative* one. Event detection starts from the alternative junctions and compares them to the canonical ones. An AS event is said to be *annotated* if the junctions of the two transcripts involved in the event are already annotated, *i.e.*, the edges representing the junctions are already present and annotated in the input graph. On the other hand, an event is said to be *novel* if the splice junctions of only one of the two transcripts are already annotated and the junctions of the other transcript are supported by read alignments only. Differently from other approaches that model AS events as coordinates over a linear reference genome, we model the AS events directly on a graph structure and we provide a formal definition of each event in terms of graph elements (i.e., vertices and edges). The annotation introduced as distinctive property of annotated spliced pangenomes helps in this formalization.

## Annotated alternative splicing events

To detect annotated alternative splicing events we consider every annotated splice junction independently as a potential event for which we check whether the following conditions apply. The definitions below are all referred to a splice junction $e = \langle a, b \rangle$ between two exonic regions having a transcript $T_1 \in \mathcal{T}(a) \cap \mathcal{T}(b)$, and the junction coverage $w(e)$ is greater than a user-specified threshold, *i.e.*, there is enough evidence for the event. In the following, given a vertex $v$, we denote with $next(v)$ and $prev(v)$ the set of successors and predecessors of $v$, respectively.

**Exon skipping.**   We call an annotated exon skipping for the annotated splice junction $j = \langle a, b \rangle$ when there is a transcript $T_2$ (of the same gene) with an exon between the two vertices $a$ and $b$—notice that $T_1 \neq T_2$ since the transcript $T_1$ uses the junction $\langle a, b \rangle$, therefore it cannot have any exon between $a$ and $b$. Formally, there exists a transcript $T_2 \in \mathcal{T}(a) \cap \mathcal{T}(b) \setminus \mathcal{T}(j)$ with two exons $e_a \in \mathcal{E}(a)$, $e_b \in \mathcal{E}(b)$ that are not consecutive in $t$. Fig 2.a shows an example in which $a = n_1$, $b = n_4$ and *T1.e2* is the skipped exon for the transcript *T2*.

**Alternative 5' splicing site.**   We call an annotated alternative 5' splicing site for the annotated junction $j = \langle a, b \rangle$ when there is another transcript (of the same gene) for which the first exon extends further towards the 3', but does not incorporate the second exon of the junction $j$. Formally, there exists a vertex $v \in next(a)$ and an exon $e \in \mathcal{E}(a) \cap \mathcal{E}(v) \setminus \mathcal{E}(b)$ (*i.e.*, the exon $e$ covers both the vertex $a$ and one of its successors $v$, but does not cover $b$) such that

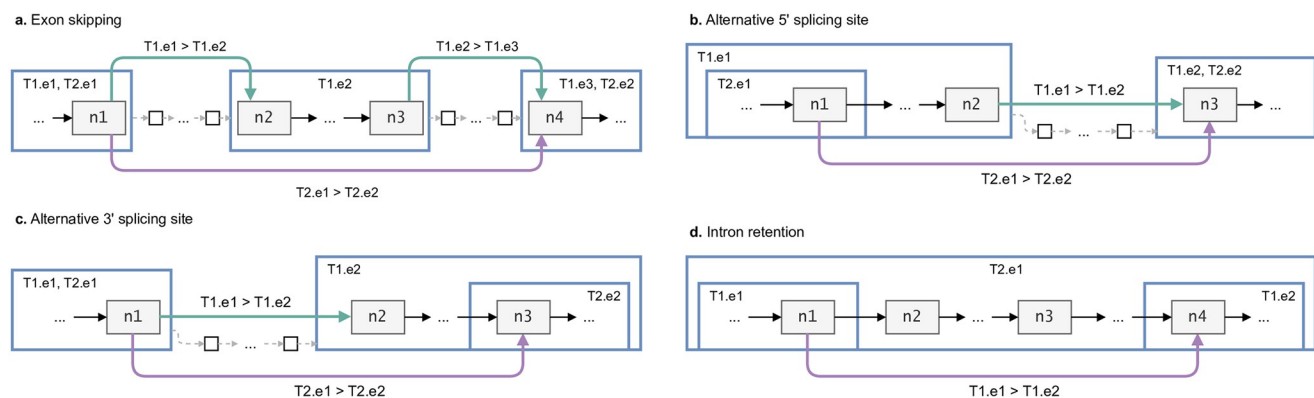

**Fig 2. Annotated events in a spliced pangenome.** All the annotated events expressed within the annotated spliced pangenome showing the different tags used: exon skipping (a), alternative donor splicing site (b), alternative acceptor splicing site (c), and intron retention (d). Haplotype information and weights are omitted for readability. Blue squares represent exons with their tags, green and purple edges are the annotated junctions with their tags, grey vertices are exonic and white are intronic. Exons are labeled with the transcript walks that are represented in the figure.

$\mathcal{T}(e) \cap \mathcal{T}(b) \neq \emptyset$. By construction, this implies that there is a transcript $T_2 \in \mathcal{T}(e)$, $T_1 \neq T_2$, that does not cover the junction $j$, but cover both exons joined by $j$. Fig 2.b shows an example in which $a = n_1$, $b = n_3$ and there are vertices $(n_1, \ldots, n_2]$ would be extending exonic vertices of *T1.e1* but not of *T2.e1*.

**Alternative 3' splicing site.** The definition of an annotated alternative 3' splicing site is symmetrical to that of 5' splicing sites. It suffices to identify a predecessor $v \in prev(b)$ and an exon $e \in \mathcal{E}(b) \cap \mathcal{E}(v) \setminus \mathcal{E}(a)$ such that $\mathcal{T}(e) \cap \mathcal{T}(b) \neq \emptyset$. Fig 2.c shows an example in which $a = n_1$, $b = n_3$ and $[n_2, \ldots, n_3)$ would be extending exonic vertices of *T1.e2* but not of *T2.e2*.

**Intron retention.** We call an annotated intron retention for the annotated junction $j = \langle a, b \rangle$ when there is another transcript (of the same gene) for which $a, b$ are in the same exon, i.e., there is a third exon spanning both exon $a$ and $b$ and including the intron in between them. Formally, there exist two edges $\langle a, u \rangle$ and $\langle v, b \rangle$ such that $\mathcal{E}(a) \cap \mathcal{E}(b) \cap \mathcal{E}(u) \cap \mathcal{E}(v) \neq \emptyset$. Notice that $u$ and $v$ are not necessarily distinct. Fig 2.d shows an example in which $a = n_1$, $b = n_4$ which are part of exons *e1* and *e2* for transcript *T1* and of *e1* for *T2*, $n_2$ and $n_3$ are intronic for *T2*.

## Novel alternative splicing events

In the definitions given below we consider an edge $e = \langle a, b \rangle$ between two vertices such that $\mathcal{T}(a) \cap \mathcal{T}(b)$ is not empty, except for the alternative 3'/5' intronic cases. This edge can be a *novel junction* if it is not annotated as such in the annotated spliced pangenome; such edges will be used to classify events by verifying the following conditions, alongside having the weight $w(e)$ greater than a user-specified threshold. Note that for every definition that requires a range of vertices $[x, \ldots y]$ to be covered, we do not always perform a complete graph traversal but we iteratively check that there exist a covered vertex $x_{i+1} \in next(x_i)$ and a covered vertex $y_{i+1} \in prev(y_i)$ for $i < w$ (for a user-specified $w$-long window) or until the sets of vertices converge. While this heuristic is less robust than an actual graph traversal, it is more computationally efficient and empirically proven to be effective.

**Exon skipping.** In this case, edge $\langle a, b \rangle$ is a novel junction and there is a transcript (of the same gene) for which none of the exons from $\mathcal{E}(a)$ and from $\mathcal{E}(b)$ are consecutive. Fig A.a in S1 Text shows an example in which $a = n_1$, $b = n_4$ which are part of exons *e1* and *e3* for transcript *T1*, meaning that according to it one exon was skipped. More formally, we call a novel

exon skipping for the novel junction $\langle a, b \rangle$ if there is at least one transcript in $\mathcal{T}(a) \cap \mathcal{T}(b)$ in which exons from $\mathcal{E}(a)$ and $\mathcal{E}(b)$ are not consecutive.

**Cassette Exon.**   We call a novel cassette exon for the junction $\langle a, b \rangle$ if it is annotated, there exist an exon from $\mathcal{E}(a)$ and one from $\mathcal{E}(b)$ that are consecutive for a transcript, and there are two novel covered edges $\langle a, x \rangle$, $\langle y, b \rangle$. Fig A.b in S1 Text shows an example in $a = n_1$, $b = n_4$ being in two consecutive exons and $x = n_2$, $y = n_3$ the intronic covered vertices in between. More formally, it holds that (i) there exist two exons $e_a \in \mathcal{E}(a), e_b \in \mathcal{E}(b)$ such that $e_a, e_b$ are consecutive for (at least) one transcript in $I$ and (ii) there are two novel covered junctions $\langle a, x \rangle, \langle y, b \rangle$.

**Alternative 5' exonic.**   We call a novel alternative 5' exonic splicing site for the novel edge $\langle a, b \rangle$ if there exist an exon from $\mathcal{E}(a)$ and one from $\mathcal{E}(b)$ that are consecutive for a transcript, there is a transcript (of the same gene) for which the first exon extends further towards the 3'. Fig A.c in S1 Text shows an example in which $a = n_1$, $b = n_3$ and $(n_1, \ldots, n_2]$ would be extending exonic vertices of $T1.e1$. More formally, it holds that (i) there exist two exons $e_a \in \mathcal{E}(a), e_b \in \mathcal{E}(b)$ such that $e_a, e_b$ are consecutive for (at least) one transcript in $I$ and (ii) $\mathcal{E}(a) \subseteq \cup_{v \in next(a)} \mathcal{E}(v)$.

**Alternative 5' intronic.**   We call a novel alternative 5' intronic splicing site for the novel edge $\langle a, b \rangle$, if $a$ is intronic and $\mathcal{E}(b)$ is exonic, there is an annotated junction $\langle x, b \rangle$ such that one exon from $\mathcal{E}(x)$ and one from $\mathcal{E}(b)$ are consecutive for a transcript, and the vertices $[x, \ldots, a]$ are covered. Fig A.d in S1 Text shows an example in which $a = n_3$, $b = n_4$, $x = n_1$ and the vertices $[n_2, \ldots, n_3]$ need to be covered. More formally, it holds that (i) $\mathcal{T}(a) \cap \mathcal{T}(b)$ is empty (ii) there exists an annotated junction $\langle x, b \rangle$, (iii) $I = \mathcal{T}(x) \cap \mathcal{T}(b)$ is not empty, (iv) there exist two exons $e_a \in \mathcal{E}(a), e_b \in \mathcal{E}(b)$ such that $e_a, e_b$ are consecutive for (at least) one transcript in $I$ and (v) each vertex in $[x, \ldots, a]$ is covered.

**Alternative 3' exonic.**   We call a novel alternative 3' exonic splicing site for the novel edge $\langle a, b \rangle$, if there is a transcript (of the same gene) for which the second exon extends further towards the 5'. Fig A.e in S1 Text shows an example in which $a = n_1$, $b = n_3$ and $[n_2, \ldots, n_3)$ would be extending exonic vertices of $T1.e2$. More formally, it holds that (i) there exist two exons $e_a \in \mathcal{E}(a), e_b \in \mathcal{E}(b)$ such that $e_a, e_b$ are consecutive for (at least) one transcript in $I$ and (ii) $\mathcal{E}(b) \subseteq \cup_{v \in prev(b)} \mathcal{E}(v)$.

**Alternative 3' intronic.**   We call a novel alternative 3' intronic splicing site for the novel edge $\langle a, b \rangle$ such that $b$ is intronic and $\mathcal{E}(a)$ is exonic, if there is an annotated junction $\langle a, x \rangle$ such that one exon from $\mathcal{E}(x)$ and one from $\mathcal{E}(a)$ are consecutive for a transcript and the vertices $[b, \ldots, x]$ are covered. Fig A.f in S1 Text shows an example in which $a = n_1$, $b = n_3$, $x = n_4$ and the vertices $[n_3, \ldots, n_4]$ need to be covered. More formally, it holds that (i) $\mathcal{T}(a) \cap \mathcal{T}(b)$ is empty (ii) there exists an annotated junction $\langle a, x \rangle$, (iii) $I = \mathcal{T}(a) \cap \mathcal{T}(x)$ is not empty, (iv) there exist two exons $e_a \in \mathcal{E}(a), e_b \in \mathcal{E}(b)$ such that $e_a, e_b$ are consecutive for (at least) one transcript in $I$ and (v) each vertex in $[b, \ldots, x]$ is covered.

**Intron retention exonic.**   We call a novel intron retention exonic for the novel edge $\langle a, b \rangle$, if $a$ and $b$ belong to the same exon, there is at least one successor of $a$ and one predecessor of $b$ that are in the same exon as $a$ and $b$. Fig A.g in S1 Text shows an example in which $a = n_1$, $b = n_4$ which are part of exons $e1$ for transcript $T1$ and $n_2$, $n_3$ would be the vertices belonging to the same exon. More formally, it holds that (i) $\mathcal{E}(a) \cap \mathcal{E}(b)$ is not empty, (ii) $\mathcal{E}(a) \cap \mathcal{E}(b) \cap \cup_{v \in next(a)} \mathcal{E}(v) \cap \cup_{e \in prev(b)} \mathcal{E}(v)$ is not empty.

**Intron retention intronic.**   Given a covered annotated junction $\langle a, b \rangle$ if the vertices $[a, \ldots, b]$ are sufficiently covered we call it an intron retention for each transcript in $\mathcal{T}(a) \cap \mathcal{T}(b)$. Fig A.h in S1 Text shows an example in which the annotated junction is $a = n_1$, $b = n_4$ and $[n_2, \ldots, n_3]$ are the intronic vertices.

## `pantas` toolkit

To quantify alternative splicing events across conditions using annotated spliced pangenomes, we developed `pantas`. The `pantas` toolkit is freely available at github.com/algolab/pantas. `pantas` takes as input a spliced pangenome in GFA format and the spliced alignments to the graph in GAF format. Since the input of `pantas` are not always readily available, we designed a Snakemake pipeline [26] that can be used to prepare the required input starting from standard file formats. The pipeline is described in Appendix A in S1 Text and is based on the `vg` toolkit, since it is the only toolkit that currently provides a spliced aligner for pangenome graphs. However, in the future, the annotation and quantification performed by `pantas` can be easily adapted to work with any other approach. We note that spliced alignments are computed against a spliced pangenome and take advantage of all the elements of the graph, e.g., a read can be aligned to any vertex of the graph (e.g., alternate alleles) and to any edge of the graph, even those not initially annotated. For instance, as shown by the alignment of "Read B" in Fig 1, a read can be aligned over the alternate allele of a variation and can also support novel splice sites by inducing novel edges not present in the graph. This allows to improve the accuracy of AS event discovery when variations occur close to splice junctions.

`pantas` starts by annotating the spliced pangenome and weighting its edges based on how many times they are used by the input alignments of each replicate. By default, `pantas` considers all alignments with alignment score $\geq 20$. This step is necessary to compute the coverage of each splice junction and successively quantify AS events. Moreover, `pantas` augments the original graph with novel splice junctions (*i.e.*, new edges) supported by the alignments and not originally present in the annotated spliced pangenome. Since a novel splice junction can start or end inside a vertex (*e.g.*, due to an alternative 5' event), we need to precisely locate the novel AS events corresponding to that junction and store such locations—an alternative way to view this novel splice junction is to split a vertex into two. Then, by analyzing the annotated spliced pangenomes that has been augmented with read alignment information, `pantas` proceeds to detect the AS events using the criteria previously described. To keep the AS detection as efficient as possible, `pantas` considers any splice junction supported by at least $w$ alignments. This is a user-defined parameter that, by default, is set to 3. A higher value of this parameter makes the computation faster but less accurate (since most events will be filtered out). From our experimental evaluations, a value of 3 resulted in a good trade-off between efficiency and accuracy. After detecting the AS events from each replicate, `pantas` quantifies the events by combining the results obtained from each replicate. `pantas` represents each AS event as a pair of sets of edges, representing the two junctions sets (the first one from the constitutive isoform and the second one from the alternative isoform) involved in the event. By analyzing the support of the junctions in each replicate, `pantas` computes the Percent Spliced-In ($\psi$) of the event by computing the ratio between the support of the constitutive isoform and the alternative isoform [27]. Once each replicate has been analyzed, `pantas` performs differential quantification of the events supported by both conditions by computing the $\Delta\psi$ of each event as the difference between the absolute value of the $\psi$ means in the two conditions.

Additionally, to simplify the downstream analysis of its results, `pantas` can also surject the positions of the splice junctions involved in the events back to a reference haplotype. Where possible, this is simply done by reporting the positions of the splice junctions w.r.t. the selected walk. This approach works well when the junctions are already annotated. In the case of novel events, `pantas` can easily surject only the junctions coming from the annotated isoforms whereas the novel junctions are not surjected. Indeed, novel AS events need more care since they usually involve novel splice sites and a novel splice junction induced by the

alignments may split a vertex of the graph in two parts. However, thanks to the information stored while augmenting the annotated spliced pangenome graph with alignment information, the correct breaking positions can be precisely identified. However, as we observed from our experimental evaluation, the lack of surjection for novel junctions does not impact the overall results of `pantas` since having at least a junction surjected to the reference haplotype allows for a more comprehensive visual inspection of the genomic locus via state-of-the-art tools, such as the Integrative Genomics Viewer [28].

## Results

To evaluate `pantas` efficacy and correctness in quantifying AS events across conditions, we performed three experimental evaluations on simulated and real data. In the first experiment, we used simulated data to evaluate the correctness of `pantas` in detecting both annotated and novel AS events. In the second experiment, we considered a real RNA-Seq dataset from Drosophila Melanogaster and we evaluated the accuracy of `pantas` in differentially quantifying annotated AS events. Finally, we evaluated the accuracy of `pantas` in quantifying RT-PCR validated AS events from a real human RNA-Seq dataset. To put our results in perspective, we compared `pantas` with three state-of-the-art approaches, namely `rMATS` [10], `SUPPA2` [14], and `whippet` [17]. The first quantifies AS events by analyzing spliced alignments to a reference genome (e.g., computed with `STAR` [29]), the second by analyzing transcript quantification computed with `salmon` [30], and the last by analyzing alignments to a custom graph, called Contiguous Splice Graph, which represents all non-overlapping exons coming from the gene annotation. We ran all tools with their default parameters and then, where indicated, we filtered the resulting set of AS events based on the reported statistical significance (as suggested by the tools' authors). However, since `pantas` does not perform any statistical validation of its events yet, in all scenarios, we considered all the events reported without any post-filtering.

### `pantas` event detection is correct and accurate

The goal of the first experimental evaluation was to assess the accuracy of `pantas` in detecting annotated and novel alternative splicing events. To do so, we simulated a RNA-Seq dataset using `asimulator` [31], a tool that simulates RNA-Seq datasets while introducing AS events. We considered the Drosophila Melanogaster reference and gene annotation (FlyBase r6.51 [32]) and the Drosophila Genetic Reference Panel (v2) [33]. As detailed in Appendix B in S1 Text, in this first experimental analysis, we considered non-overlapping genes from the gene annotation and single nucleotide polymorphisms from the reference panel. Not dealing with indels allowed us to use the state-of-the-art `asimulator` for reads and events simulation without requiring additional (and more error-prone) steps. Future works will be devoted to the development of a full-fledged pangenomic simulator for RNA-Seq reads and AS events. To simulate a real-case scenario, where the sequenced sample is typically not present in the reference panel, we randomly selected a sample from the panel, we simulated reads from both its haplotypes, and then removed the considered sample from the reference panel. We used this reduced panel while preparing the input for `pantas`. We ran the 4 tools (`pantas`, `rMATS`, `SUPPA2`, and `whippet`) with their default parameters and we evaluated their accuracy in terms of Precision, Recall, and F1-Measure computed by comparing the true events reported by `asimulator` with the events reported by the tools. Since `asimulator` reports the simulation locus for each read, we used this information to compute the true support of each AS event, i.e., the number of reads supporting the junctions involved in the event. We then created several truth sets varying the minimum support value $\mathcal{W}$ of the events and we computed the

accuracy of the tools at each point. In order to avoid penalizing the tools' precision for calling an event that is actually a true event but supported by less than $\mathcal{W}$ reads, we computed the false positives of each tools with respect to the original truth set and we considered the $\mathcal{W}$-filtered truth sets when computing true positives and false negatives. More details on the simulation settings can be found in Appendix B in S1 Text. Fig 3 reports the results of this analysis on annotated events.

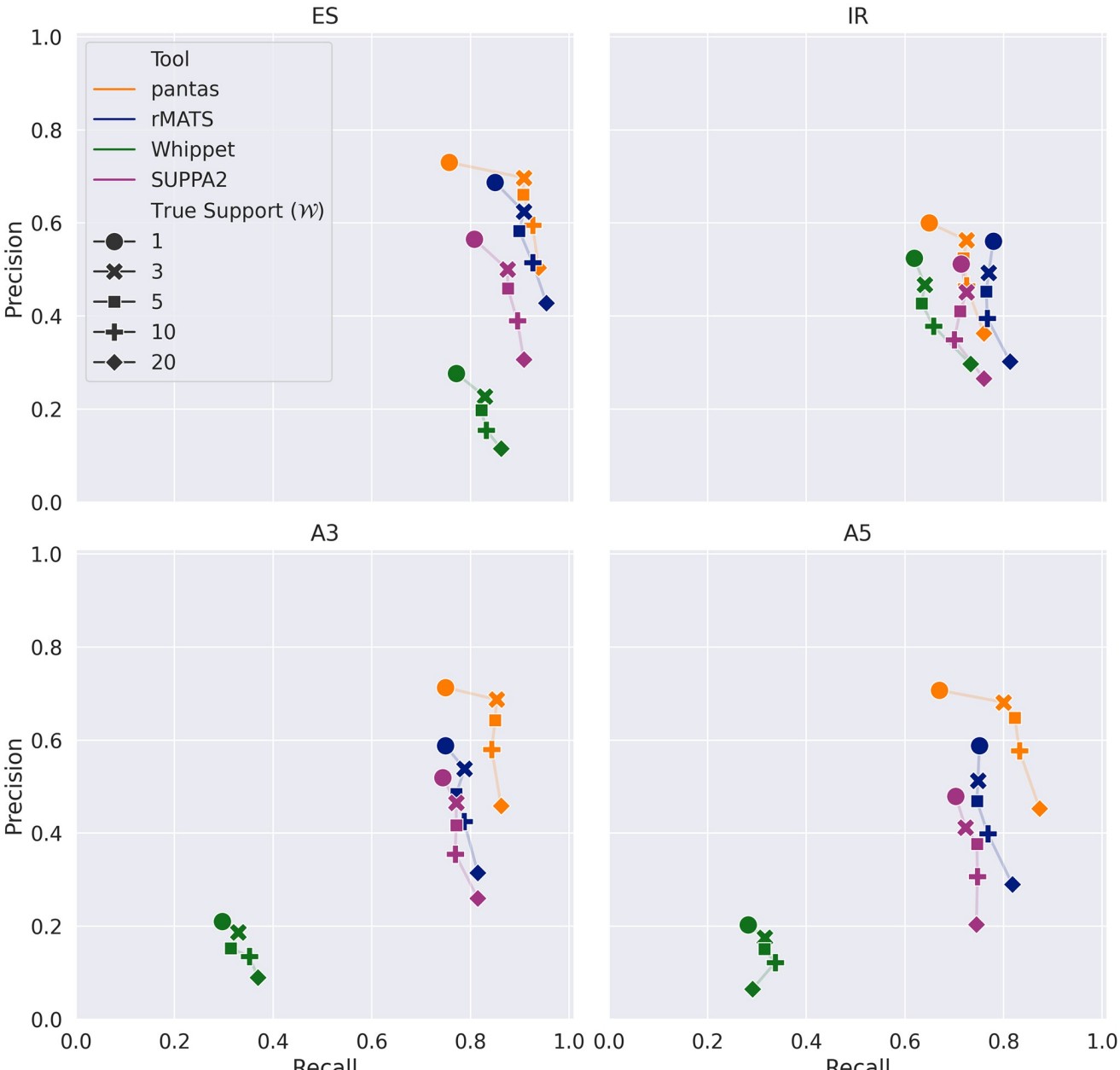

**Fig 3. Results on simulated data from Drosophila Melanogaster (annotated events).** Precision and recall are computed by comparing `asimulator` truth (filtered based on $\mathcal{W}$, that is the minimum number of reads supporting an event) with the output of each tool. Results are broken down by event type (ES: Exon Skipping, IR: Intron Retention, A3: Alternative 3', A5: Alternative 5').

As expected, when considering all events reported by `asimulator` without any filtering on their true support (i.e., $\mathcal{W} = 1$), all tools achieved higher precision at the expense of their recall. Filtering out lowly supported events allowed to increase the recall of the tools while lowering their precision. Indeed, as reported in Table A in S1 Text), a higher threshold $\mathcal{W}$ resulted in a smaller truth set of events (hence in a lower number of true positives). The increase of performance is more pronounced for `pantas` than for other tools, especially when moving from $\mathcal{W} = 1$ to 3. This is due to the lack of a statistical validation of `pantas` results. Indeed, we recall that `pantas` does not perform any statistical validation of its events and its calling step is mainly guided by the parameter $w$: `pantas` considers all junctions with at least $w$ alignments (3, by default) supporting it. This matches the results presented in Fig 3: when we consider all events reported by `asimulator` with coverage at least 3, `pantas` achieved the best accuracy results for almost all event types. Only on intron retentions, `pantas` did not manage to achieve the same recall of the other best approach (`rMATS`). However, it always achieved the best precision. As reported in Table B in S1 Text, when $\mathcal{W} = 3$, `pantas` achieved a recall of 0.725 (-0.045 w.r.t. `rMATS`) but a precision of 0.563 (+0.07 w.r.t. `rMATS`), making it the most accurate approach, with an F1 of 0.634. Overall, in the annotated event setting, when considering all events with good support ($\mathcal{W} > 1$), `pantas` and `rMATS` resulted the most accurate tools for detecting alternative splicing events, followed by `SUPPA2`. We notice that in this analysis we were interested in evaluating the capability of each tool in detecting AS events and not in quantifying them. Therefore, we did not perform any post-filtering of the events reported by the tools based on their statistical significance (when available). Such a post-filtering would have improved the precision of the tools while lowering their recall.

In the novel event scenario, instead, we could not include `SUPPA2` and `whippet` in the analysis. Indeed, `SUPPA2` can detect only annotated events. On the other hand, `whippet` allows for augmenting the Contiguous Splice Graph with novel splice sites supported by read alignments, but we did not manage to run this augmentation successfully: `whippet` crashed when we asked to include novel splice sites from a BAM file. Without novel splice sites, `whippet` has been able to detect only novel exon skipping events, hence we decided to not include it in this analysis. Fig B and Table C in S1 Text report the results of the analysis on novel events. As expected, on novel events, due to the higher complexity of the task of detecting events not already present in the given annotation, the two tools achieved lower accuracy. However, the results follow the same trend of the results on annotated events, with `pantas` achieving the best trade-off between precision and recall when the true support threshold is set to 3. Differently from annotated events, `pantas` did not manage to reach the accuracy of `rMATS` when calling exon skipping events (-0.014 loss in F1 when $\mathcal{W} = 3$) while it managed to achieve the best precision and recall for all other event types, especially when the true events are highly supported. Remarkably, in this setting, `pantas` resulted the most accurate approach for detecting novel intron retentions, which are the hardest events to detect correctly [34]. Overall, the good accuracy achieved by `pantas` in both annotated and novel scenarios proves its validity and the efficacy of its precise formalization of alternative splicing events on annotated spliced pangenomes.

Even though the accuracy of `pantas` heavily depends on the chosen value of its $w$ parameter, these results show that, when properly set, it enables accurate AS event detection. Indeed, when the $w$ parameter is properly set to match the expected coverage of the events, `pantas` achieved the best trade-off between precision and recall. However, even when this parameter does not match the expected coverage, `pantas` has proved to be still able to achieve competitive (if not superior) accuracy (especially for highly supported events). To test this claim, we reran all the analyses increasing the $w$ parameter to 5 (results for annotated and novel events reported in Fig C and D in S1 Text, respectively). As expected, this time `pantas` managed to

achieve the best trade-off between precision and recall when the true event support ($\mathcal{W}$) is $\geq 5$. However, setting this parameter is not easy since RNA-Seq coverage is not always uniform. Anyway, we believe that the default value (3) should be used in most of the scenarios. Future efforts will be focused in removing this dependency on the $w$ parameter by including a statistical validation of `pantas` results.

### `pantas` quantification shows good correlation with state-of-the-art

In the second part of our experiments, we evaluated the quality of `pantas` differential quantification of AS events from real RNA-Seq data. To this aim, we considered a recent study [35] that performed a genome wide transcriptome analysis of the ageing process in Drosophila Melanogaster (SRA BioProject ID: `PRJNA718442`). We considered the FlyBase (r6.51) genome and gene annotation and the SNPs and indels from the Drosophila Genetic Reference Panel (v2). In this analysis, we considered only non-overlapping genes to allow for a more precise comparison. Indeed, overlapping genes increase the noise in the output of all tools and add further complexity in the AS quantification step since usually short RNA-Seq alignments in these regions is not sufficient to discern the different transcripts (coming from different genes) expressed in the dataset. Differently from the previous experimental evaluation, where we used the default simplification provided by `pantas`, in this analysis, to reduce the complexity of spliced pangenome indexing, we opted for its more aggressive simplification which can alter complex haplotype-aware transcripts to reduce graph complexity. Nevertheless, we had to perform a subsampling of the input RNA-Seq dataset and we randomly extracted 1/3 of the reads (approximately 8 750 000) from each replicate using `seqtk`. We have not been able to consider the entire samples since `vg mpmap` required more than 3 days to align a single replicate (using 32 threads). This time requirement made any analysis infeasible. Unfortunately, `vg mpmap` is currently the only approach available to perform spliced alignment against a spliced pangenome, but this does not restrict `pantas` usability, which has been devised and developed as a general approach that can potentially analyze the spliced alignments to a spliced pangenome computed by any spliced aligner. Overall, `pantas` (without taking into account the input preparation step) completed its differential analysis in less than 2 hours using 32 threads and required 16GB of RAM, making it a practical solution. Extended results on the efficiency of `pantas` and other tools are reported in Table F in S1 Text.

As done in our previous experiment, we ran all tools with their default parameters. Since in this scenario we used real data without any wet-lab validation, we could not compare the results of the tools with a ground-truth but we performed an all-vs-all comparison of the $\Delta\psi$ provided by the 4 tools (`pantas`, `rMATS`, `whippet`, and `SUPPA2`). We considered only annotated events (in order to include all tools) and we removed all events without a strong evidence of differential change between the two conditions, i.e., events with $|\Delta\psi| \geq 0.05$. Moreover, since most approaches compute the significance of differential splicing by performing tests on read counts supporting the event junctions, we also filtered out from the resulting set of events all those events reported as *non statistically significant* by the tool (when available). We considered all events reported by `rMATS` and `SUPPA2` with a p-value strictly smaller than 0.05 and all events reported by `whippet` with a probability greater or equal to 0.9. As shown in Fig 4a, the resulting set of events considered in our analysis consists in 1396 events for `pantas`, 932 events for `rMATS`, 4322 events for `whippet`, and 513 events for `SUPPA2` (Table D in S1 Text reports these numbers broken down by event type). Out of these events, 164 are shared among the 4 tools. When comparing the quantification ($\Delta\psi$) reported by the tools on this subset of events, we noticed a strong correlation (Fig 4b). `pantas` achieved very high correlation with `rMATS` (Pearson correlation: 0.948) and `whippet` (0.921) while lower

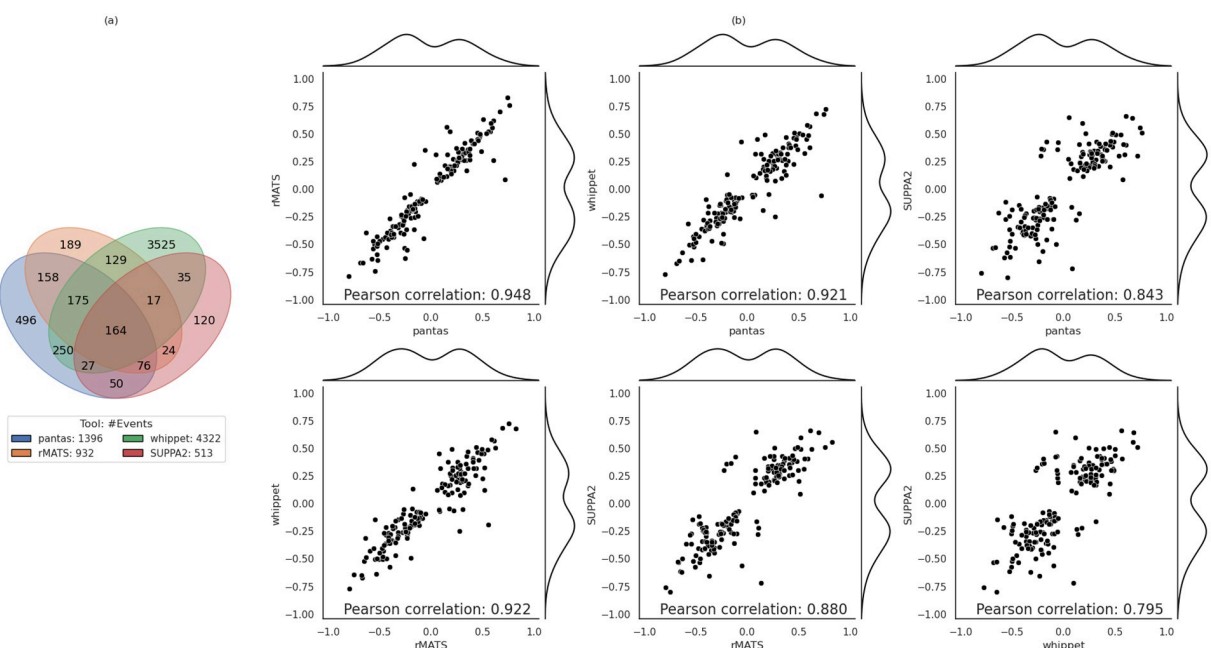

**Fig 4. Results on real data from Drosophila Melanogaster.** (a) Venn diagram showing the number of AS events reported by each tool. (b) All-vs-all correlation plots of the $\Delta\psi$ reported by the considered tools for the 164 events shared among the tools. Detailed version where each point is color coded based on the event type is available at Fig E in S1 Text.

correlation with SUPPA2 (0.843). Remarkably, pantas, rMATS, and whippet achieved higher correlation when compared among them but lower correlation when compared with SUPPA2. This was somehow expected since SUPPA2 quantifies AS events starting from transcript quantification (based on $k$-mer analysis) and not from read alignment. As expected, considering all events reported by the tools, i.e., without any post-filtering on the statistical significance (when available), leads to an increase in the number of events (Fig F.a and Table E in in S1 Text) and a strong loss of correlation (Fig F.b in S1 Text). However, also in this scenario, pantas and rMATS have been able to achieve the strongest correlation (0.832). The correlation achieved by pantas and all other approaches based on read alignment proves that read alignment to spliced pangenome is effective and can be confidently used for AS event differential quantification.

## pantas is effective in quantifying RT-PCR AS events

In the last part of our experimental evaluation, we considered a real human RNA-Seq dataset and we assessed the performance of pantas in detecting and quantifying RT-PCR validated AS events. Similarly to some recent studies [12, 14, 17], we analyzed the RNA-Seq dataset provided by [36] (SRA BioProject ID: PRJNA255099) and we evaluated pantas accuracy in quantifying RT-PCR validated events. The RNA-Seq dataset consists of three replicates for two conditions (control condition versus double knockdown of the TRA2 splicing regulatory proteins, TRA2A and TRA2B). We considered the human genome and gene annotation from Ensembl [37] (release 109) and the 1000 Human Project phase 3 VCF files [38]. We considered 128 random samples from the EUR superpopulation, arbitrarily chosen, i.e., 128 individuals of European ancestry (for a total of 256 haplotypes encoded in the graph, which is almost three times the amount of haplotypes provided by the Human Pangenome Reference Consortium

[1])) and all variations (SNPs and indels) with at least one alternate allele expressed in the considered population. Since this analysis focuses on a small panel of genes (77 RT-PCR validated genes), in order to reduce the time complexity of the entire `pantas` pipeline, we constructed a spliced pangenome limited to the genomic loci of interest. We did this by using the third simplification methodology provided by `pantas` (described in Appendix A in S1 Text). This procedure is especially effective when the user is interested in analyzing a panel of genes, that is a common scenario in transcriptomics [39]. This *gene-centric* pipeline is provided alongside the main `pantas` code base as a companion approach that can be used by the user when the focus of the analysis is a set of genes. We note that we did not manage to build the full human spliced pangenome `GCSA2` index due to the limited amount of RAM available on our server (256GB). For this reason, we decided to adopt the alternative procedure we developed to analyze a panel of genes. In any case, this does not limit the applicability of `pantas` since, when a more powerful infrastructure is available, it can be effectively used to analyze the full spliced pangenome. By retaining only the genes of interest, we managed to reduce the times of the entire pipeline to less than 3 hours and the RAM requirements to less than 8GB, without degrading the AS events detection accuracy. Detailed results are reported in Table G in S1 Text. However, aligning a full input RNA-Seq dataset against a reduced graph deteriorates the mapping accuracy (reads may be placed on the wrong gene since the correct gene is not in the graph) and the running times (the aligner tries to find the best spot for all the reads, also those not coming from the genes of interest). For this reason, we decided to filter the input RNA-Seq dataset using `shark` [39]. This step retains only those reads that potentially come from the genes of interest. Since this preprocessing may alter the quantification, we used it only with `pantas`. All other tools (`rMATS`, `SUPPA2`, `whippet`) have been run on the original inputs, without any preprocessing. Anyway, as shown in [39] and as proven by our results, this preprocessing based on accurate read filtering does not affect the quantification step.

In this scenario, we ran all tools with their default parameters. We evaluated each tool accuracy in terms of correctly detected events w.r.t. the RT-PCR validated events, then we compared the predicted $\Delta\psi$ with the experimental $\Delta\psi$ provided by RT-PCR validation. Our evaluation focused on 77 RT-PCR validated exon skipping events (details on preprocessing of the truthset can be found in Appendix C in S1 Text). Fig 5 reports the results of our analysis when considering only events reported as statistically significant by the tools (using the same criteria applied

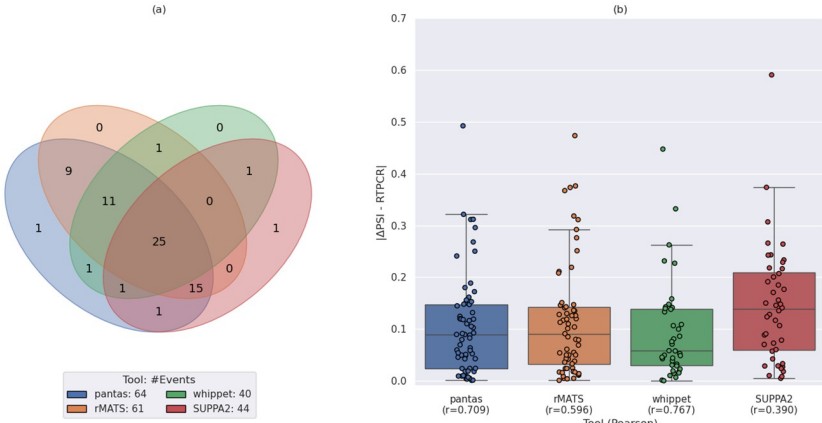

**Fig 5. Results on real data from human.** (a) Venn diagram showing the number of statistically significant differential AS events reported by each tool. The legend reports the total number of events reported by the tool. (b) Boxplot showing the distribution of the difference between the $\Delta\psi$ predicted by each tool and the $\Delta\psi$ provided by RT-PCR. The x axis labels report the tool name and the Pearson correlation ($r$) between the predicted $\Delta\psi$ and the RT-PCR $\Delta\psi$. Note that since `pantas` do not compute statistical significance, we filtered only by the reported $\Delta\psi$.

in our previous experiment on real data from Drosophila Melanogaster) and with an absolute magnitude of change greater than 0.05 between the two conditions, i.e., events with $|\Delta\psi| \geq 0.05$. Since `pantas` does not perform statistical validation, we filtered its set of events based only on the reported $\Delta\psi$. Fig G in S1 Text reports the same results obtained without any filtering. Regarding the number of events detected by each tool, `pantas` reported the highest number of events (64 out of 77 RT-PCR validated events), followed by `rMATS` (61), `SUPPA2` (44), and `whippet` (40). Notice that `SUPPA2` and `whippet` reported fewer events since 14 out of the 77 RT-PCR events are novel events and, as previously shown, they cannot correctly detect novel events. Remarkably, only 25 events were reported by all 4 tools, 1 event was only reported by `pantas` whereas 3 events were missed by our approach. The reason behind these results needs to be sought in the filtering step we performed. Indeed, as shown in Fig G.a in S1 Text, when considering all events, each tool reported a higher number of events (as expected) and the number of events reported by all 4 tools increased to 43. When removing `whippet` from this analysis, the number of events reported by `pantas`, `rMATS`, and `SUPPA2` increased even more (61), showing good agreement among the three tools. Moreover, in this setting no event was reported by `pantas` only. However, 9 events remained undetected by `pantas`. By manually investigating both `STAR` and `vg mpmap` alignments over the loci of these events, we noticed that the two aligners show good agreement: the skipping junction of all events show no support in the control replicates and very small support (1 or 2 alignments) in the knockdown replicates. The same holds for the 3 events missed by only `pantas` in the filtered analysis. In this case, 2 events show no support in the control replicates while 1 events has been reported with $\Delta\psi =$ 0.04. A more polished statistical analysis of `pantas` results might allow to recover these events. Surprisingly, when considering the filtered set of events reported by the tools, 10 RT-PCR events were not reported by any tool. On the other hand, when considering all events reported by the tools, all RT-PCR validated events were detected.

We then evaluated the correctness of the quantification provided by each tool by comparing the predicted $\Delta\psi$ with the experimental $\Delta\psi$ provided by the RT-PCR validation. As shown in Fig 5, `whippet` quantification showed the best correlation with the RT-PCR expected quantification (Pearson correlation: 0.767) followed by `pantas` (0.709). Anyway, all the correlations are weak. Moreover, `whippet` reported fewer events than `pantas` (40 against 64). Remarkably, although the distributions of the differences between the RT-PCR $\Delta\psi$ and the quantifications provided by `pantas` and `rMATS` are very similar (with a 0.01 average difference advantage for `pantas`), `rMATS` showed lower correlation than `pantas`. However, as previously reported [14], all tools struggle in achieving a good correlation with the RT-PCR quantification. The same trend can be observed when considering all events reported by the tools (Fig G.b in S1 Text). Remarkably, in this scenario, the correlation between `pantas` quantification and RT-PCR quantification increased and reached the correlation achieved by `whippet`. This indicates that a more statically robust confidence score for `pantas` quantification is an interesting and compelling focus for future efforts. Finally, as done in [14], we evaluated the false positive rate of each tool using 44 RT-PCR negative exon skipping events that did not show any relevant change between the two conditions. Remarkably, `pantas` reported only one false positive, followed by `SUPPA2` (28), `whippet` (32), and `rMATS` (37). When filtering the sets of events reported by the tools using the aforementioned criteria, `pantas` did no report any false event. `SUPPA2` reported 2 false events, followed by `whippet` (4) and `rMATS` (10). In conclusion, although `pantas` statistical validation is still not fully polished, the differential quantification computed from alignments to an annotated spliced pangenome is robust: `pantas` is able to retrieve a good amount of RT-PCR validated events without introducing false calls. Overall, the results described in this section suggest a

strong agreement between the different approaches for the differential quantification of AS events with no method clearly outperforming the others.

## Discussion

The recent adoption of a reference pangenome for analyzing genetic variability in humans opens the perspective of building a pantranscriptome that provides a complete picture of gene annotation in human population. However, moving from a reference-based gene annotation to a haplotype-aware gene annotation, where genes are annotated w.r.t. multiple genomes, as recently proposed in [24], requires to rethink tools for transcriptomic analysis, similarly to what has been done in computational graph pangenomics for analyzing genomic variants in a population. In this paper, we advanced the investigation started in [2] of spliced pangenomes by proposing `pantas`, the first pangenomic approach for detecting and quantifying AS events across RNA-Seq conditions. Our approach is based on the novel notion of annotated spliced pangenome, introduced here as an enhanced version of a spliced pangenome, and on the precise formalization of AS events on this structure. An extensive experimental evaluation on simulated data shows that `pantas` achieves higher accuracy than competitors, also thanks to its formal definitions of AS events on a graph structure. On real data, `pantas` obtains results that are comparable to those of state-of-the-art methods based on a linear reference, while being able to take into account the genetic variability of the population under investigation. In the experiments carried out in this paper, we considered haplotype-aware gene annotations, where genes are annotated w.r.t. individual genomes of a population. `pantas` exploits variants information to obtain more accurate read alignments and to improve AS predictions. An interesting future direction consists in analyzing the impact of haplotype-aware annotation on reference bias reduction while performing AS events quantification. `pantas` can be the right tool to perform such an analysis. However, the definition of haplotype-aware annotations is still not settled and many interesting questions are still unanswered. For instance, if an in-frame insertion on a haplotype leads to a longer exon, it is still not clear if the longer exon needs to be classified as a new exon or an alternative to the "reference" one. This also affects AS events detection and it may require the introduction of new definitions, such as those of haplotype-specific AS events. Currently, `pantas` adopts the "classic" reference-based AS events definitions and extend them to work with haplotype-aware annotations. However, we observe that those definitions are not "haplotype-specific" yet, since they might mix transcripts coming from different individuals and do not take into account complex scenarios such that those previously described.

Future work will focus on extending `pantas` to cyclic graphs in order to take into account more complex chromosomal rearrangements and on incorporating more complex AS events (such as mutually exclusive exons) and Local Splicing Variations [15, 16]. The main limitation of `pantas` is its dependency on external tools for the preparation of the required input. The heavy computational load required by `vg index` to index an annotated spliced pangenome and the long running times of the `vg mpmap` spliced aligner potentially limit the applicability of `pantas`. To solve this problem, we designed an alternative strategy that reduces the computational complexity without affecting the accuracy of `pantas`. This new procedure is especially effective when the analysis focuses on a panel of genes. We plan to focus future research efforts on the development of novel approaches for building and indexing a spliced pangenome as well as for mapping RNA-Seq reads to the graph, with the goal of reducing the computational requirements. Compelling alternatives to the current index for spliced pangenomes can be sought in repeat-aware indices for collections of linear strings, such as the r-index [40, 41] and the Movi index [42]. The extension of these indexes to spliced pangenomes or the

development of new indices will benefit `pantas` and any future tool for the analysis of a pantranscriptome.

## Supporting information

**S1 Text. Supplementary material.** Appendices, Supplementary Figures, and Supplementary Tables. Appendix A, "Spliced pangenome construction and alignment". Appendix B, "Details on the simulation settings". Appendix C, "RT-PCR truthset filtering". Fig A: novel events definitions on annotated spliced pangenome. Fig B: results on simulated data from Drosophila Melanogaster (novel events, default parameters). Fig C: results on simulated data from Drosophila Melanogaster (annotated events, $w = 5$ for `pantas`). Fig D: results on simulated data from Drosophila Melanogaster (novel events, $w = 5$ for `pantas`). Fig E: results on real data from Drosophila Melanogaster. Fig F: results on real data from Drosophila Melanogaster (all events). Fig G: results on real data from Human (all events). Table A: number of simulated events from Drosophila Melanogaster. Table B: full results on simulated data from Drosophila Melanogaster (annotated events). Table C: full results on simulated data from Drosophila Melanogaster (novel events). Table D: events count for Drosophila Melanogaster experiments on real data. Table E: events count for Drosophila Melanogaster experiments on real data (all events). Table F: efficiency results on real data from Drosophila Melanogaster. Table G: efficiency results on real Human data.
(PDF)

## Author Contributions

**Conceptualization:** Gianluca Della Vedova, Luca Denti.

**Data curation:** Simone Ciccolella, Davide Cozzi, Luca Denti.

**Investigation:** Gianluca Della Vedova, Stephen Njuguna Kuria, Luca Denti.

**Methodology:** Simone Ciccolella, Davide Cozzi, Stephen Njuguna Kuria, Luca Denti.

**Software:** Simone Ciccolella, Davide Cozzi, Luca Denti.

**Supervision:** Gianluca Della Vedova, Paola Bonizzoni, Luca Denti.

**Validation:** Simone Ciccolella, Davide Cozzi, Luca Denti.

**Writing – original draft:** Simone Ciccolella, Luca Denti.

**Writing – review & editing:** Simone Ciccolella, Davide Cozzi, Gianluca Della Vedova, Paola Bonizzoni, Luca Denti.

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
