## [Decision Letter · Decision Letter 0]

11 Jun 2024

Dear Dr Denti,

Thank you very much for submitting your manuscript "Differential quantification of alternative splicing events on spliced pangenome graphs" for consideration at PLOS Computational Biology.

As with all papers reviewed by the journal, your manuscript was reviewed by members of the editorial board and by several independent reviewers. In light of the reviews (below this email), we would like to invite the resubmission of a significantly-revised version that takes into account the reviewers' comments.

Please pay attention to the comments on data, code and reproducibility. Per PLoS data policy the code for reproducing and replicating the analysis must be provided.

We cannot make any decision about publication until we have seen the revised manuscript and your response to the reviewers' comments. Your revised manuscript is also likely to be sent to reviewers for further evaluation.

Sincerely,

Wei Li, Ph.D.

Academic Editor

PLOS Computational Biology

Shihua Zhang

Section Editor

PLOS Computational Biology

Please pay attention to the comments on data, code and reproducibility. Per PLoS data policy the code for reproducing and replicating the analysis must be provided.

Reviewer's Responses to Questions

**Comments to the Authors:**

Reviewer #1: Ciccolella et al. describes the pandas tool to detect alternative splicing events using annotated spliced pangenomes. This is the first pangenomic method for AS detection. Even though the idea itself is interesting, the manuscript could be improved by providing more evidences and examples to demonstrate the advantage of this new method.

In section 2.1, what is the formal definition of a “walk”? Was the annotated spliced pangenome constructed by the authors, or readily available from existing resources? If it was constructed by the authors, is it made publicly available to the community?

In section 2.2, what is the tolerance level to consider a splicing event novel? If a splice junction only differs from an annotated one by 1 bp, is it still considered novel?

After reading the whole manuscript, it is not clear what the exact output of pantas look like. Can the authors provide a more specific description with examples?

Is pandas only designed for short reads? Can it be used with long reads?

The manuscript compares pantas with three existing tools and all three tools are able to evaluate statistical significance. I wonder why pantas does not perform statistical validation using similar methods as those used by other tools.

Why didn’t the author compare pantas with leafcutter, which is also a popular tool?

The current simulation is based on Drosophila genome. What if the simulation is done using the human genome, which may have more complex splicing events? Additionally, can the authors use simulation to evaluate how method performance is impacted by sequencing depth?

“As done in our previous experiment, we ran all tools with their default parameters, except for pantas, for which we set the minimum junction support to 5, in order to achieve a good trade-off between true AS events and false calls.” This approach leads to the question that how users could select this parameter in their own analysis to achieve a good tradeoff?

“An extensive experimental evaluation shows that pantas achieves results that are comparable with those of state-of-the-art methods based on a linear reference”. Can the authors clarify when it is advantageous to use pantas other than state-of-the-art methods?

The manuscript describes how to prepare input files for pantas in supplementary sections. It will be helpful to incorporate these steps in the github tutorial with example datasets.

Reviewer #2: Review uploaded as an attachment

**Have the authors made all data and (if applicable) computational code underlying the findings in their manuscript fully available?**

Reviewer #1: **No: **There is a github repository for the software. However, the code for reproducing the analyses in the manuscript is not available.

Reviewer #2: **No: **Some summary statistics (such as average runtimes) do not have the backing data points included in the paper, and it is not clear how to obtain them beyond re-running the authors' experimental code. It is also not clear what kind of computer specifically it would need to be re-run on to replicate them.

PLOS authors have the option to publish the peer review history of their article (what does this mean?). If published, this will include your full peer review and any attached files.

Reviewer #1: No

Reviewer #2: No
---

## [Decision Letter · Decision Letter 1]

17 Oct 2024

Dear Dr Denti,

Thank you very much for submitting your manuscript "Differential quantification of alternative splicing events on spliced pangenome graphs" for consideration at PLOS Computational Biology. As with all papers reviewed by the journal, your manuscript was reviewed by members of the editorial board and by several independent reviewers. The reviewers appreciated the attention to an important topic. Based on the reviews, we are likely to accept this manuscript for publication, providing that you modify the manuscript according to the review recommendations.

Sincerely,

Wei Li, Ph.D.

Academic Editor

PLOS Computational Biology

Shihua Zhang

Section Editor

PLOS Computational Biology

Reviewer's Responses to Questions

**Comments to the Authors:**

Reviewer #1: The revision has successfully addressed all my questions.

Reviewer #2: The authors have substantially revised both their software and their manuscript in light of the reviewers' comments. The pantas tool now implements a defensible version of what alternative splicing detection could be like in a pangenomic context. While it could still be improved with the addition of built-in statistical testing, the rewritten manuscript no longer abuses the word "significant" and more defensibly places adding statistical testing under future work. The new analysis of the simulation experiments, with its precision-recall curves, is noticeably easier to understand than the previous tables, and the discussion of read simulation makes it much more clear that ignoring indels in the simulation is a load-bearing part of the procedure. The revised discussion of the RT-PCR concordance analysis also makes more sense.

My only remaining major concern is the way that the authors filtered events from the truth set in the simulated-data analysis for Figure 3. It looks like the authors are taking the set of events produced by each tool, and the set of events in each filtered truth set, and calling any event produced by the tool but not in the *filtered* truth set a "false positive". So, when running pantas with its w parameter set to 3, on the filtered truth set filtered at ω=20, if pantas sees an event that asimulator actually simulated, that has a read support of 10, and detects and produces it in its output, it gets penalized for creating a "false positive", since the event did not have a read support of at least 20. I don't think that this is a reasonable way to do this analysis. I also think that this approach is going to over-represent the performance of pantas, when one of the conditions tested has pantas's w parameter equal to the truth set filtering parameter ω; in Figure 3 this is at w = ω = 3. In that condition, pantas has access to the information that no event with a read support of less than 3 should be emitted, and will suppress any events it thinks have less read support even if it thinks they exist. The other tools don't have access to the true minimum read support, and will produce events where they think the read support is less than 3, because they don't know such events are ineligible to be in the truth set even if they were actually simulated into the data. This will differentially increase the false positive rate of tools other than pantas.

The authors should instead take an approach more typical of precision-recall or ROC plot curves, and keep the truth set constant along the curve while thresholding the output of each tool at different levels. This is how variant calling evaluation tools like hap.py work, keeping the set of true variants constant and filtering the call set under evaluation at different quality/confidence thresholds. If all tools report some notion of the observed read support for an event, then the events from each tool could be thresholded on that to produce the different points along each curve. Alternatively, a tool-specific measure of confidence could be used to get a curve for each tool.

An alternate approach might be to identify all the true positives, false positives, and false negatives between each tool's results and the full, unfiltered truth set. Then, for each minimum read support value ω, drop the true positives and false negatives involving truth set events with *true* read support less than ω. This would avoid penalizing tools' computed precision scores when the tools are "too good" at identifying true events with low support, while still penalizing them for identifying false events with low support, and it would not be necessary for each tool to report a confidence or observed read support value.

Either approach would avoid the circularity of having conditions where pantas has access to the true minimum allowed read support while other tools do not.

In addition to this one major point, I also found the following minor issues:

* The manuscript still contains a lot of number agreement errors and other similar copy-editing problems; the editors should go over it closely to find and remove these.

* The visual difference between italic w, the pantas parameter, and lower case omega, the truth set filtering threshold, is very slight. It might be good to rename one of these symbols, or typeset one of them in a different font.

* The author summary contains the text "Pange3PM ESTnome", which is presumably a typo.

* The authors write that "pantas can pave the way to next generation bioinformatic approaches for the accurate analysis of the relations between genetic variations and alternative splicing aberrations". This seems, firstly, too specific: pantas could be useful for analysing relationships between genetic variation and all forms of alternative splicing, not just "aberrations". It also sounds vaguely eugenic: if a person can have genetic variation that causes alternative splicing aberrations, it is easy to slip into thinking of those alleles as aberrant alleles and that person as a person whose genes are substandard. Given the existence of non-scientists eager to find scientific work that they can deliberately misconstrue as supporting racism, as described in e.g. https://www.nature.com/articles/d41586-022-03252-z, it is important to write defensively around these issues. It does seem like "aberrant splicing events" is a concept in the field, defined along the lines of "rare large alterations of splice isoform usage" in https://doi.org/10.1038/s41588-023-01373-3, but if the authors are only going to use the word once in the summary, without expanding on what they mean, it might be safest to choose a different word. Perhaps this could be just "between genetic variations and alternative splicing", or "between genetic variations and medically-relevant alternative splicing".

* The text in Figure 1 is much too small to read when viewing the figure at a normal size on a normal screen.

* The precision/recall plots in Figure 3 probably should be square, with equal scales on both axes, to avoid visually prioritizing either precision or recall over the other metric.

* The Figure 3 legend for "True Support" is in ascending order of support threshold imposed on the truth set, and so starts with the condition where tools are expected to achieve the highest precision and proceeds to conditions where the expected precision is lower. But this is right to left along the X axis in the plots, and backward from how I want to read them. The legend for the point shapes should be flipped so that it describes the points in the order they are expected to appear left to right.

* None of the Venn diagrams (like Figure 4a) are scaled to visually show the number of items in each category. They would be more useful if they were. The authors might also consider using UpSet plots here instead of Venn diagrams, which could turn these into more visually readable bar charts.

* In section 3.2, "an increase in the number of events (Suppl. Fig. S6b and Suppl.Table S5)" should reference S6a, and "strong loss of correlation (Suppl. Fig. S6a)" should reference S6b; the panel letters seem to be swapped.

* In section 3.3 the authors write that they built a spliced pangenome from "128 random Europen samples" from the 1000 Genomes Project. I think "Europen" was probably meant to be "European". But that still doesn't make sense: the 1000 Genomes data doesn't categorize any samples as "European". It *does* have an "EUR" superpopulation that the data collectors created, described on their data portal as "European Ancestry", but that superpopulation includes many samples not properly describable as "European" due to not being sampled in Europe, such as "Utah residents (CEPH) with Northern and Western European ancestry". There are also a number of European samples, such as those from "Indian Telugu in the U.K.", that were sampled in Europe but not assigned to the EUR superpopulation. If the authors really drew from samples annotated as belonging to the EUR superpopulation in the 1000 Genomes VCF, then they need to say that, because it is an importantly different thing to be assigned to that superpopulation than it is to be European. If they drew from all samples collected in Europe, they should say that more explicitly.

* The authors don't justify drawing from only whatever they consider to be "Europen" samples, rather than from the whole 1000 Genomes dataset. Why was this decision made? It seems like maybe they are hoping to ethnicity-match or ancestry-match the MDA-MB-231 cell line from their reference 6. But that sample isn't from a 1000 Genomes population (which would generally require something like 4 grandparents with very similar self-reported ancestry). It instead comes with the outdated and vague (see https://www.nature.com/articles/d41586-021-02288-x) descriptor of "Caucasian" (see https://www.cellosaurus.org/CVCL_0062), and was apparently collected at M D Anderson in Texas, which isn't in Europe. Since a racial descriptor like this isn't useful for ruling out recent ancestry from places other than Europe, or from populations that would be placed in 1000 Genomes superpopulations other than EUR, what justifies the restriction on sampling? There doesn't seem to actually be enough known about the ancestry of the target sample to try and match it, or to justify assuming that restricting sampling like this will produce better and not worse results. Was this particular restriction on sampling empirically discovered to produce better results than unrestricted sampling? Is there an analysis of the cell line that the authors could cite that shows that it looks a lot like the EUR superpopulation as a whole, but also not specifically like one or a few of the individual populations in it (which would then be better sets to sample from)?

* For the 44 RT-PCR negative differential alternative splicing events, 2/44 significant hits is nearly exactly the expected number of false discoveries for a p=0.05 threshold, when presented with 44 candidates from the null model's distribution. It might make sense to note this when describing the results from for the tools that perform statistical significance testing; on their own terms they *shouldn't* be producing noticeably fewer false positives than that.

**Have the authors made all data and (if applicable) computational code underlying the findings in their manuscript fully available?**

Reviewer #1: None

Reviewer #2: Yes

PLOS authors have the option to publish the peer review history of their article (what does this mean?). If published, this will include your full peer review and any attached files.

Reviewer #1: No

Reviewer #2: No

Figure Files:

Data Requirements:

Reproducibility:

References:

---

## [Decision Letter · Decision Letter 2]

21 Nov 2024

Dear Dr Denti,

We are pleased to inform you that your manuscript 'Differential quantification of alternative splicing events on spliced pangenome graphs' has been provisionally accepted for publication in PLOS Computational Biology.

Best regards,

Wei Li, Ph.D.

Academic Editor

PLOS Computational Biology

Shihua Zhang

Section Editor

PLOS Computational Biology

Feilim Mac Gabhann

Editor-in-Chief

PLOS Computational Biology

Jason Papin

Editor-in-Chief

PLOS Computational Biology

Reviewer's Responses to Questions

**Comments to the Authors:**

Reviewer #2: The authors have addressed or acknowledged all of my concerns. I'm slightly worried that saying that samples are drawn "from the EUR superpopulation, arbitrarily chosen," is interpretable as saying that the samples themselves are chosen arbitrarily (and not randomly), but I think that's the last thing.

**Have the authors made all data and (if applicable) computational code underlying the findings in their manuscript fully available?**

Reviewer #2: Yes

PLOS authors have the option to publish the peer review history of their article (what does this mean?). If published, this will include your full peer review and any attached files.

Reviewer #2: No

---

## [Editor Report · Acceptance letter]

29 Nov 2024

PCOMPBIOL-D-24-00606R2 

Differential quantification of alternative splicing events on spliced pangenome graphs

Dear Dr Denti,

I am pleased to inform you that your manuscript has been formally accepted for publication in PLOS Computational Biology. Your manuscript is now with our production department and you will be notified of the publication date in due course.

With kind regards,

Zsofia Freund
